# Expression of PDLIM5 Spliceosomes and Regulatory Functions on Myogenesis in Pigs

**DOI:** 10.3390/cells13080720

**Published:** 2024-04-21

**Authors:** Yu Fu, Shixin Li, Jingru Nie, Dawei Yan, Bo Zhang, Xin Hao, Hao Zhang

**Affiliations:** 1National Engineering Laboratory for Livestock and Poultry Breeding, Beijing Key Laboratory of Animal Genetic Engineering, China Agricultural University, Beijing 100193, China; fuyu6486@163.com (Y.F.); 13508635177@163.com (S.L.); 13627606200@163.com (J.N.); bozhang0606@cau.edu.cn (B.Z.); 2College of Animal Science and Technology, Yunnan Agricultural University, Kunming 650201, China; 1995027@ynau.edu.cn

**Keywords:** pig, *PDLIM5*, muscle development, single nucleotide polymorphisms, promoter activity

## Abstract

Meat yield, determined by muscle growth and development, is an important economic trait for the swine industry and a focus of research in animal genetics and breeding. PDZ and LIM domain 5 (PDLIM5) are cytoskeleton-related proteins that play key roles in various tissues and cells. These proteins have multiple isoforms, primarily categorized as short (PDLIM5-short) and long (PDLIM5-long) types, distinguished by the absence and presence of an LIM domain, respectively. However, the expression patterns of swine PDLIM5 isoforms and their regulation during porcine skeletal muscle development remain largely unexplored. We observed that PDLIM5-long was expressed at very low levels in pig muscles and that PDLIM5-short and total PDLIM5 were highly expressed in the muscles of slow-growing pigs, suggesting that PDLIM5-short, the dominant transcript in pigs, is associated with a slow rate of muscle growth. PDLIM5-short suppressed myoblast proliferation and myogenic differentiation in vitro. We also identified two single nucleotide polymorphisms (−258 A > T and −191 T > G) in the 5′ flanking region of PDLIM5, which influenced the activity of the promoter and were associated with muscle growth rate in pigs. In summary, we demonstrated that PDLIM5-short negatively regulates myoblast proliferation and differentiation, providing a theoretical basis for improving pig breeding programs.

## 1. Introduction

During embryogenesis, skeletal muscles in vertebrate limbs develop from muscle progenitor cells derived from somites [1]. Postnatal muscle growth primarily relies on the expansion of muscle fiber cell volume. Muscle development involves a series of biological processes, including myoblast proliferation, differentiation, and fusion, leading to the formation of multinuclear myotubes, which mature into functional muscle fibers [2]. The growth of these muscle fibers is primarily regulated by a range of specific muscle-related transcription factors [3] and protein kinases [4]. Myogenic regulatory factors serve as pivotal hubs within the gene transcription network linked to muscle development, directly orchestrating gene expression to promote skeletal muscle development [5]. Cyclin-dependent kinases regulate the muscle cell cycle progression [6,7,8]. The precise timing and spatial location of the expression of these regulators are strictly controlled to ensure the accurate and orderly progression of muscle development [9,10].

The cytoskeleton plays a key role in maintaining the internal structure of cells, bearing external forces, and preserving cell morphology. Cytoskeleton-related proteins play an indispensable role in cytoskeletal changes. PDZ and LIM domain 5 (PDLIM5) are cytoskeleton-associated proteins belonging to the PDZ and LIM domain families, originally discovered using yeast two-hybrid screening methods [11]. PDLIM5, a scaffolding protein, comprises one N-terminal PDZ domain and three C-terminal LIM domains, which facilitate protein–protein interactions [12]. *PDLIM5* generates multiple splice variants conserved across humans [13], rats [14], and mice [15]. Human *PDLIM5* contains 9 isoforms (PDLIM5 isoform a–i). The long *PDLIM5* containing LIM (*PDLIM5-long*) is widely expressed in various tissues, whereas short *PDLIM5* without the LIM domain (*PDLIM5-short*) is specifically expressed in cardiac and skeletal muscles [16]. The expression of different spliceosomes occurs at distinct temporal points [17,18,19]. *PDLIM5-long* is predominant in the embryonic heart of mice, and the short isoforms are abundant in the adult heart [17] and skeletal muscles [18]. Comparable findings were also reported in the human myocardium [19], indicating that *PDLIM5* fulfills distinct roles in different tissues. The porcine *PDLIM5* encodes 11 isoforms; Among these, isoforms X1-9 possess a long LIM domain and are classified as *PDLIM5-long*, and X10 and X11 less LIM domain are classified as *PDLIM5-short*.

PDLIM5 binds to cytoskeletal and membrane proteins via its PDZ domain and interacts with various signaling molecules, including protein kinases and transcription factors, through its LIM domain [20]. Consequently, PDLIM5 plays diverse roles in various tissues and cells, including those of the heart, skeletal muscle, kidney [21], and nervous system [22,23,24]. PDLIM5 interacts with CREB to promote cardiomyocyte growth [25], and *PDLIM5*-knockout mice exhibit defects in heart contraction [15]. Additionally, *PDLIM5* can play a role in the development of various diseases, including cancer [26,27,28], pulmonary hypertension [29], depressive disorders [30], and type 2 diabetes and hypertension [31], highlighting the significance of exploring the function of *PDLIM5*. The short and long isoforms of *PDLIM5* seem to play distinct roles and have distinct functions in the development of striated muscles [18]. The expression of the long isoform of PDLIM5 gradually decreases during early myoblast differentiation and postnatal heart development [18], and it may facilitate the activation of satellite cells and the myogenic differentiation of C2C12 cells by targeting the inhibitor of DNA binding 1 or 2 (Id1 or Id2) [32,33]. The promotion of long splicing variants during myogenesis is also conserved in chicken skeletal muscle satellite cells [34]. Conversely, the expression of *PDLIM5-short* increases in the late postnatal period and restricts cardiomyocyte cytokinesis, hypertrophy, and myogenic differentiation [18], indicating a potential negative regulatory effect of *PDLIM5-short* in striated muscle development. However, the function and expression patterns of *PDLIM5* with different splicing isoforms in pigs remain largely unknown. Our previous study based on transcriptome and proteome profiles demonstrated that *PDLIM5* is differentially expressed in pigs exhibiting divergent growth phenotypes [35]. Therefore, we hypothesized that *PDLIM5*, a candidate gene for growth traits, exhibits differential expression in pigs and regulates muscle growth.

This study investigated the tissue expression profiles of swine *PDLIM5* isoforms and identified their functions in myoblasts proliferation and differentiation. Additionally, we screened for single nucleotide polymorphisms (SNPs) across multiple pig breeds and analyzed the association between *PDLIM5* and growth traits in these pigs.

## 2. Materials and Methods

### 2.1. Experimental Materials

Tibetan (TP) and Wujin (WJ) pigs, native Chinese pigs with slow growth characteristics, and large white (LW) pigs, fast-growing commercial pigs introduced from abroad, were reared at the Tibet Agriculture and Animal Husbandry College. For RNA extraction, at least six embryos from each group (TP, WJ and LW) were sampled from two pregnant sows after 60 days of insemination. The heart, back fat (BF), lung, liver, leg and intestinal embryonic tissues were used, all of which were used for semi-quantitative reverse transcriptional polymerase chain reaction (sqRT-PCR). Additionally, embryonic longissimus dorsi (LD) muscle tissues from the 12th rib were sampled for both sqRT-PCR and quantitative real-time polymerase chain reaction (qRT-PCR). Diannan small-ear (DSE) and New Huai populations (NHP), which are native Chinese pigs, were reared in Xishuangbanna, Yunnan and Dingyuan, Anhui, respectively. Ear samples were collected from four breeds (DSE, *n* = 16; TP, *n* = 34; LW, *n* = 24; NPH, *n* = 97) for DNA extraction. The animal feeding and tests were conducted based on the National Research Council Guide for the Care and Use of Laboratory Animals and were approved by the Institutional Animal Care and Use Committee (approval number: AW80203202-1-1).

### 2.2. RNA Extraction and Expression Analysis

RNA was extracted using TRIzol reagent (Invitrogen, Carlsbad, CA, USA) according to the manufacturer’s instructions. Subsequently, the RNA was reverse-transcribed into cDNA by using the FastKing One Step Reverse Transcription Kit (TIANGEN, Beijing, China). Target gene expression was evaluated using sqRT-PCR and qRT-PCR. qRT-PCR was performed using SYBR Green Master Mix (TIANGEN) and specific primers (Appendix A). Gene expression was analyzed using the 2^−ΔΔCt^ method [36]. Target genes expression was normalized to *GAPDH* expression, which was used as a control.

### 2.3. SNP Screening, Genotyping, and Correlation Analysis

The primers for SNP screening of PDLIM5 (NC_010450.4) are listed in Appendix A. In general, the sequence approximately 2000 bp upstream of the gene is commonly cloned as the promoter of the gene. Thus, the amplification sequences covered approximately 2000 bp upstream of the PDLIM5 transcription start site. SNPs among different pig populations were screened and identified using PCR and Sanger sequencing (ABI3030XL DNA, Thermo Fisher Scientific, Wilmington, DE, USA). In brief, the PCR products amplified from ten individuals of each breed were pooled, and the purified products were sequenced to identify the SNPs by using Sanger sequencing. Chromas Pro (Technelysium, South Brisbane, Australia) and DNAMAN6.0 (Lynnon, San Ramon, CA, USA) were used to analyze and interpret the sequencing results. For determining the genotypes of the identified SNPs, individual samples were separately used as templates for PCR amplification, and the purified products were sequenced by using Sanger sequencing. Chromas Pro and DNAMAN6.0 software were used to analyze the peak patterns to determine the genotype. Ninety-seven boars, individuals of age weighing 30 and 90 kgs, were obtained from the Anhui Kexin Pig Breeding Farm.

### 2.4. Dual-Luciferase Reporter Assay

The *PDLIM5* promoter region was cloned into the PGL3 luciferase reporter vector (Addgene, Cambridge, MA, USA). Then, the *PDLIM5* promoter luciferase construct was transfected into cells in a 24-well plate for 48 h. Subsequently, the cells were lysed using 100 μL lysis buffer, and promoter activity was assayed using a dual luciferase reporter assay system (Promega, Madison, WI, USA). The luciferase activities were measured using a PerkinElmer 2030 Multilabel Reader (PerkinElmer, Waltham, MA, USA). The primers used are listed in Appendix A.

### 2.5. Primary Myoblasts Isolation and Culture

Primary myoblasts were isolated from the LD muscle of pigs. In brief, the minced muscles were digested using 0.2% type II collagenase (Sigma-Aldrich, St. Louis, MO, USA) and 2.5 U/mL dispase (Roche Applied Science, Nutley, NJ, USA) for 1 h. The filtered cell suspension was centrifuged at 1000× *g* for 5 min and cultured in Ham’s F10 nutrient mixture medium (Gibco, Waltham, MA USA), which was supplemented with 20% fetal bovine serum (Gibco), 1% penicillin-streptomycin (Gibco), and 5 ng/mL basic fibroblast growth factor (PeproTech, Cranbury, NJ, USA). The cells were incubated in an atmosphere of 5% CO_2_ atmosphere at 37 °C. DMEM supplemented with 2% horse serum (HS; Gibco) was used to induce myogenic differentiation.

### 2.6. Plasmid Construction and Transfection

The full-length sequence of porcine *PDLIM5* was amplified using full-length-F/R primers (F: 5′-tagcgtttaaacttaATGAGCAACTACAGTGTGTC-3′; R: 5′-atccgagctcggtacCTGTACGTTAAGAGCACGTG-3′) and cloned into the pcDNA3.1 plasmid. For functional study, myoblasts were transfected with *PDLIM5* overexpression plasmid using lipofectamine 3000 (Invitrogen, Carlsbad, CA, USA) according to the manufacturer’s instructions.

### 2.7. Cell Counting Kit-8 (CCK8) and EdU Assays

After transfection for 24 h, cells were incubated with CCK8 (10 µL reagent dissolved in 100 µL DMEM, Beyotime Biotechnology, Shanghai, China) at 37 °C for 1–4 h in the dark. The absorbance at 450 nm was measured using a microplate reader (Biotek, Winooski, VT, USA) to determine the proliferation ability. For 5-ethynyl-2-deoxyuridine (EdU, Ribobio, Guangzhou, China) assays, cells were seeded in 96-well plates and transfected with control or overexpression plasmid for 24 h. Subsequently, transfected cells were incubated with 50 μM EdU at 37 °C for 2 h. Finally, the proliferated cells labeled using EdU were imaged under a microscope (Leica, Heidelberg, Germany).

### 2.8. Immunofluorescence

Immunofluorescence was performed according to our published method [37]. In brief, differentiated cells were fixed using 4% paraformaldehyde and permeabilized with 0.5% Triton X-100 for 15 min. After blocking for 1–2 h, the cells were incubated overnight with a mouse anti-myosin heavy chain antibody (cat. no. M4276, Sigma-Aldrich, St. Louis, MO, USA; 1:500) at 4 °C. Subsequently, the cells were washed thrice with PBS and incubated with a fluorescently labeled secondary antibody (A11032; Thermo Fisher Scientific, Wilmington, DE, USA, 1:400) for 1 h in the dark. The nuclei were stained with DAPI (Roche Applied Science, Nutley, NJ, USA) for 5 min. Digital images were captured using a fluorescence microscope (Leica image analysis system, Model Q500MC, Leica, Heidelberg, Germany).

### 2.9. Statistical Analysis

Statistical analyses were performed using one-way ANOVA and Student’s *t*-test. Results are presented as mean ± standard deviation (SD). Differences were considered statistically significant at *p* < 0.05. A χ2 test was used to analyze the distribution of genotypes and compare the differences in genotype distribution.

## 3. Results

### 3.1. Expression and Identification of Main PDLIM5 Isoforms in Pigs

*PDLIM5* has diverse splice variants, which may display different expression patterns and functions. *PDLIM5-short* was highly expressed in the heart, LD, and leg muscles in pigs, and *PDLIM5-long* was expressed at extremely low levels in various porcine tissues, including muscles (Figure 1A). In addition, total *PDLIM5* (encoding all splice variants) was detected in the LD tissues of pigs with different growth rates. Slow-growing TP and WJ pigs exhibited significantly higher *PDLIM5* expression in the LD tissues than fast-growing LW pigs (Figure 1B), which was consistent with our previous transcriptomic results [35]. Similarly, *PDLIM5-short* exhibited the same expression trend as total *PDLIM5* (Figure 1C). These results indicate that *PDLIM5-short* is the dominant transcript in pigs and is likely associated with muscle growth rate. Consequently, we further focused on *PDLIM5-short* and conductedstudies on its potential role.

### 3.2. Inhibition of Inhibited Myoblast Proliferation by PDLIM5

As *PDLIM5-short* is the main splice variant in pigs, a *PDLIM5-short* overexpression plasmid was constructed and transfected into porcine primary myoblasts to explore its biological functions during myogenesis (Figure 2A). *PDLIM5* overexpression markedly reduced the absorbance value after CCK8 treatment (Figure 2B) and EdU positivity, compared with that of the control (Figure 2C). Overexpression of *PDLIM5* effectively downregulated proliferative genes, including cyclin D1 (*CCND1*), *Ki67*, proliferating cell nuclear antigen (*PCNA*), and cyclin-dependent kinase 4 (*CDK4*) (Figure 2D–G), further highlighting the negative role of *PDLIM5* in myoblast proliferation.

### 3.3. PDLIM5 Suppressed Myogenic Differentiation

We investigated the role of *PDLIM5* in porcine myoblasts during various stages of differentiation. Overexpression of *PDLIM5* significantly inhibited myoblast differentiation, as demonstrated by decreased transcript levels of myogenin (*MyoG*), myogenic differentiation (*MyoD*), and myosin heavy chain (*MyHC)* on the second and fourth day of differentiation (Figure 3A–C). To confirm these results and visually observed myotube differentiation, we performed myosin immunofluorescence staining. Consistent with our expectations, fewer myotubes were formed by myoblasts overexpressing *PDLIM5* than by those transfected with a control vector (Figure 3D,E). This findingindicates that *PDLIM5* suppresses the myogenic differentiation of porcine myoblasts.

### 3.4. Screening of SNP Sites in the PDLIM5 Gene

To identify promoter regions and SNPs that may affect transcription, we transfected C2C12 cells with different lengths of the *PDLIM5* promoter region. The results showed that the region from −521 to +86 bp upstream of the translation start site (+1) of the promoter regions exhibited the strongest relative luciferase activity, indicating that this region served as the core promoter of *PDLIM5* (Figure 4A).

Two SNPs, −258 A > T, and −191 T > G, were identified in the core promoter region of *PDLIM5*. The genotype and allele frequencies of these two SNPs in different breeds are listed in Table 1. In LW, the locus of −258 A > T was the AA genotype, and the site in mini-type indigenous breeds (DSE and TP) exhibited three genotypes (AA, AT, and TT) in Hardy–Weinberg Equilibrium (*p* > 0.05). The −191 T > G site showed higher frequencies of allele T in the introduced breeds (LW, frequencies of 0.938) than in indigenous pigs (DSE and TP; frequencies of 0.844 and 0.265 respectively). Collectively, the genotypes of the LW populations at both loci were almost all AT/AT, and different mutated genotypes were observed in the DSE and TP populations. To verify the effects of different genotypes at these two sites on promoter activity, we amplified and cloned four haplotype sequences formed by the combination of the two sites into dual-luciferase vectors (Figure 4B). The transcriptional activity of the promoters was significantly different, (the order being TG > AG > TT > AT) (Figure 4C).

The NHP population exhibited three genotypes at each SNP locus (Table 2). We compared the ages of the 30 and 90 kg body weights among the three genotypes. At the −258 A > T site, the AA genotype was associated with significantly faster growth than the AT and TT genotypes (*p* < 0.01). The TT genotype at the −191 T > G site showed faster growth than the TG and GG genotypes (*p* < 0.01). A further joint analysis of these two loci revealed that six combined genotypes were present in the NHP population (TG/TG = 3, AT/AG = 9, AT/AT =69, AT/TT = 5, AT/TG = 10, and AG/TG = 1), and the growth rate of AT/AT-type pigs was significantly higher than that of the other groups (Table 3).

## 4. Discussion

*PDLIM5* contains the PDZ and LIM domains, which are associated with the cytoskeleton [38] and kinases [39]. We found that *PDLIM5* was highly expressed in the muscles of slow-growing pig breeds, consistent with our prior study [35], suggesting its potential role in muscle development. *PDLIM5* can be categorized into long isoforms with LIM domains, and short isoforms without LIM domains [20]. Long- and short-splice isoforms exist in mammals, including humans [16] and mice [14]. The splicing transition from LIM-containing PDLIM5-long to LIM-free *PDLIM5-short* was reported during postnatal heart development and in the early period of C2C12 differentiation [18]. In this study, we explored the role of two isoforms of the *PDLIM5* gene in the regulation of skeletal muscle development in pigs. *PDLIM5-long* showed very low expression in pig muscles. *PDLIM5-short*, as the main isoform, was highly expressed in the muscles of slow-growing pigs, and its expression pattern was similar to that of total *PDLIM5*. These findings correlate the relative presence of the different isoforms with muscle growth rate, suggesting that the role of alternative splicing variants of the same gene has different effect on skeletal muscle biology. Especially, *PDLIM5* short isoforms without LIM may have important but unknown functions in mature striated muscles. Therefore, we explored the function of *PDLIM5-short* in porcine muscle cells by using gene overexpression.

PDLIM5 acts as a scaffold that controls diverse cellular functions by interacting with various molecules [26]. *PDLIM5* prevents proliferation and cell cycle progression in the nervous system [40]. Gan et al. reported that the RNA binding protein with multiple splicing represses *PDLIM5-short* isoforms by splicing exon 8 of *PDLIM5*, maintaining the expression of *PDLIM5-long* variants in embryonic cardiomyocytes. The absence of RBPMS leads to the accumulation of *PDLIM5-short* variants, which arrest the cell cycle and disrupt cardiomyocyte cytokinesis [41]. This results are consistent with our finding that *PDLIM5-short* inhibits myoblast proliferation. *PDLIM5-short* overexpression effectively downregulated the level of proliferative genes, including *Ki67*, *PCNA* and *CDK4*, but there was no significant decrease observed in the *CCND1* level. This phenomenon may involve multiple complex factors. First, the expression of *CCND1* is not solely controlled by a single gene but is part of a complex network involving multiple upstream genes and transcription factors. The overexpression effect of *PDLIM5-short* may be counteracted or modified by other regulatory factors. Second, gene expression is regulated at multiple levels, including transcriptional regulation, post-transcriptional modification, protein translation, and modification. Therefore, even if a gene is overexpressed at the transcriptional level, its ultimate impact on *CCND1* expression may be limited by other regulatory mechanisms. Furthermore, intracellular gene expression is a dynamic equilibrium process with intricate interactions and feedback mechanisms among various genes and proteins. Overexpression of *PDLIM5-short* may trigger a cascade of reactions that may not be directly correlated with *CCND1* expression.

The different functions of *PDLIM5-long* and *PDLIM5-short* in myoblast differentiation and hypertrophy were also identified. A study proved that *PDLIM5-long* promoted mice satellite cell activation during myogenesis [32]. However, *PDLIM5-short* disrupted normal muscle differentiation [42]. In rat neonatal cardiomyocytes, overexpression of the *PDLIM5-long* isoform increased the expression of hypertrophy markers and cell size, and the overexpression of the *PDLIM5-short* isoform prevented these hypertrophic changes [17]. Similarly, we observed that overexpression of LIM-free *PDLIM5* significantly inhibited the expression of differentiation markers and myotube formation in porcine myoblasts, revealing a negative regulatory effect of *PDLIM5-short* isoform on muscle differentiation in pigs. This results also indicates that *PDLIM5* is functionally conserved in mammalian muscle development.

These results show that *PDLIM5-short* may inhibit the growth and development of muscles by inhibiting myoblast proliferation and differentiation. This phenomenon is consistent with earlier results showing high expression of *PDLIM5-short* in native pig breeds with slow growth rates. Yamazaki et al. reported that *PDLIM5-long* promoted the C2C12 myogenic differentiation through the repression of Id2 translocation from the cytoplasm to the nucleus [25]. *PDLIM5-short* variants act as antagonists of *PDLIM5-long* variants in postnatal mouse hearts [17]. Regarding whether *PDLIM5-short* can also act as an antagonist of the *PDLIM5-long* variant in muscle cells to affect protein nuclear translocation, or whether *PDLIM5-short*, the main splicing variant in pigs, has a unique but unknown mechanism of regulating myoblast proliferation and differentiation, these questions require further experimental verification.

A study reported that polymorphisms in *PDLIM5* are associated with an increased risk of dilated cardiomyopathy [43]. In this study, we performed segmented amplification of 2000 bp upstream of the transcription start site of the *PDLIM5* gene and found that the promoter activity showed an increasing trend with the truncation of the fragment length. Thus, we speculated that the promoter region of *PDLIM5* contains transcription suppression sites. The promoter activity in the −521 to +86 bp region was the highest, and we identified two SNPs (−258 A > T and −191 T > G) that formed four haplotypes (AT, AG, TT, and TG) in this region. The promoter activity of haplotype TG was significantly higher than that of AT, and TG haplotypes were mainly distributed in TP pigs, and almost all of haplotypes were AT in LW pigs. This results seemed to explain why the expression level of *PDLIM5* was higher in TP pigs than in LW pigs. Additionally, correlation analysis of gene polymorphisms and growth rate showed that genotype AA of the −258 A > T site, genotype TT of the −191 T > G locus, and combined genotype AT/AT of the two loci were associated with fast body growth within NHP populations. Therefore, our results suggest that these two SNPs regulate the expression of *PDLIM5* and postnatal body growth in pigs and are potential molecular markers for use in pig breeding. The regulation of gene expression and muscle development by SNPs requires further research in porcine primary myoblasts.

## 5. Conclusions

Our study revealed that *PDLIM5-short* was the dominant isoform in the muscle tissues of pigs and was highly expressed in the muscles of slow-growing pigs and that *PDLIM5* inhibited myoblast proliferation and myogenic differentiation. We explored the biological functions of *PDLIM5* and identified SNPs associated with the muscle growth rate that were used for genetic improvement. This findings provide a basis for further research on the mechanisms underlying muscle development and molecular breeding in pigs.

## Figures and Tables

**Figure 1 cells-13-00720-f001:**
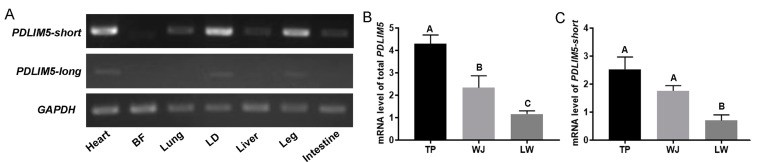
Characterization of *PDLIM5* expression in pigs. (**A**) Determination of *PDLIM5* expression in different tissues of TP pigs at the embryonic stage using sqRT-PCR. Expression of total PDLIM5 (**B**) and PDLIM5-short (**C**) at the transcript level in the LD of three pig breeds. Total PDLIM5 contains all splice variants; PDLIM5-short, short isoform of PDLIM5 without the LIM domain, and PDLIM5-long, long isoform PDLIM5 with the LIM domain. LD for longissimus dorsi; BF for back fat. TP for tibetan pig (*n* = 6), WJ for wujin pig (*n* = 6), LW for large white (*n* = 6). Each bar represents the means ± SD. Different letters indicate significant differences between groups.

**Figure 2 cells-13-00720-f002:**
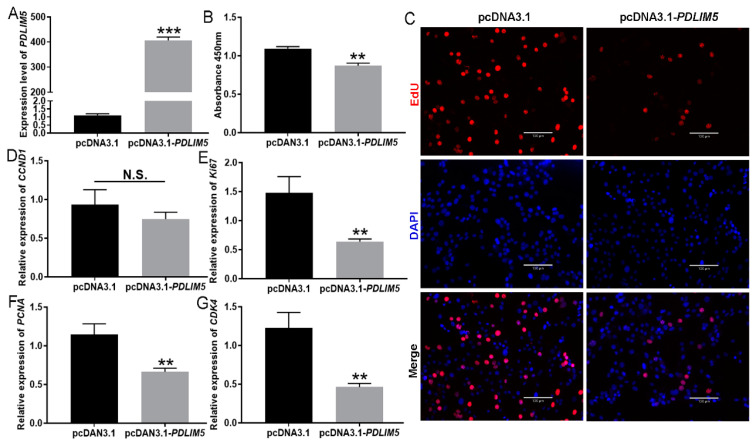
*PDLIM5* inhibits myoblast proliferation. (**A**) Efficiency of the detection of *PDLIM5-short* overexpression plasmid. (**B**) CCK8 assay of proliferating myoblasts transfected with overexpression constructs. (**C**) Representative images of EdU staining for proliferated cells after pcDNA3.1-*PDLIM5* transfection. Blue indicates nuclei stained with DAPI, red indicates EdU-positive proliferating cells, scale bar = 130 µm. (**D**–**G**) The mRNA expression levels of proliferative genes. *GAPDH* is used as a reference gene. The data represent the mean ± SD of three independent experiments, ** *p* < 0.01, *** *p* < 0.001, N.S., not significant.

**Figure 3 cells-13-00720-f003:**
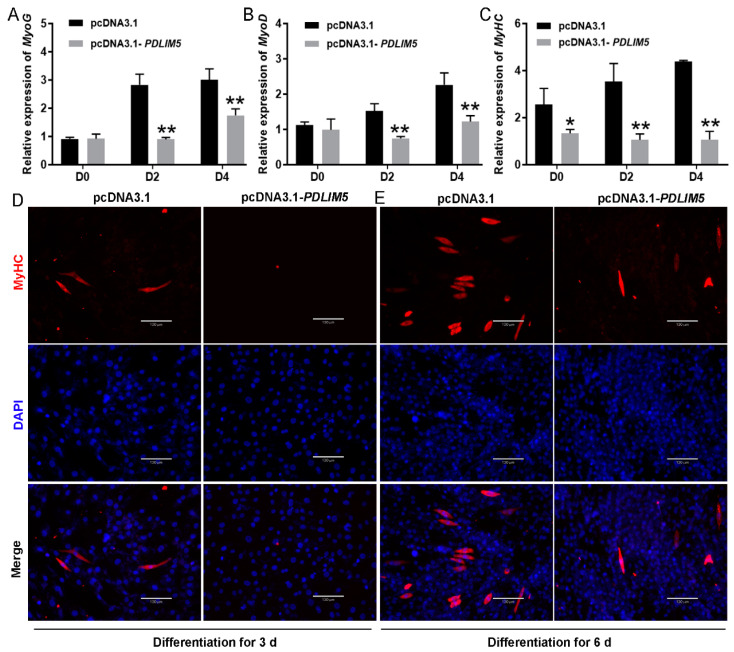
*PDLIM5* suppresses myogenic differentiation. (**A**–**C**) Expression of myogenic genes (*MyoG*, *MyoD*, and *MyHC*) at the transcript level. Porcine myoblasts were transfected with pcDNA3.1 or *PDLIM5-short* overexpression plasmid, and then induced to differentiate for 0, 2, and 4 d (D0, D2, and D4). (**D**,**E**) Representative images of MyHC staining for myoblasts after pcDNA3.1-*PDLIM5* transfection. Transfected cells cultured with differentiation medium for 3 and 6 d. Blue indicates nuclei stained with DAPI; red indicates MyHC-positive myotubes; scale bar = 130 µm. The data represent the mean ± SD of three independent experiments. *GAPDH* has been used as a reference gene. * *p* < 0.05, ** *p* < 0.01.

**Figure 4 cells-13-00720-f004:**
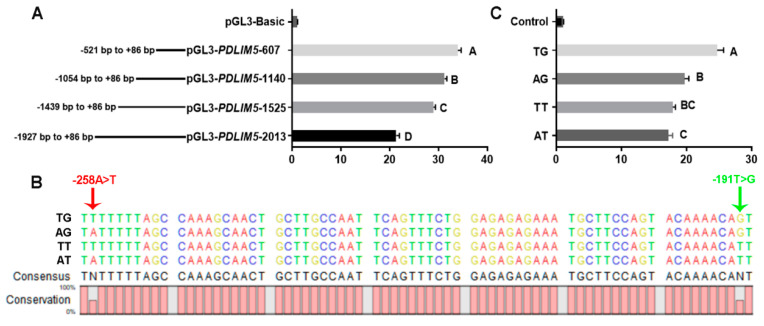
SNP sites and promoter activity analysis. (**A**) Comparison of the activities of double luciferase vectors expressed after transfection of C2C12 cells with different fragment lengths. (**B**) Sequencing results of −521 bp ± 86 bp double luciferase active vectors constructed with four different haplotypes (TG, AG, TT, and AT). Arrows indicate the sites of two SNPs, −258 A > T (red) and −191 T > G (green). (**C**) Dual-luciferase analysis for promoter activity of four haplotype double luciferase vectors in −521 bp ± 86 bp. Each bar represents the mean ± SD of six independent experiments. Different capital letters represent significant differences, *p* < 0.01.

**Table 1 cells-13-00720-t001:** Gene and genotype frequencies in −258 A > T and −191 T > G in different pig breeds.

Loci	Breed	Sample Size	Genotype Frequency			Allele Frequency		χ2 Value (*p*-Value)
−258 A > T			AA	AT	TT	A	T	
	DSE	16	0.688	0.187	0.125	0.781	0.219	0.881 (0.664)
	TP	34	0.471	0.353	0.176	0.647	0.353	1.105 (0.576)
	LW	24	1	0	0	1	0	/
−191 T > G			TT	TG	GG	T	G	
	DSE	16	0.750	0.188	0.062	0.844	0.156	1.155 (0.561)
	TP	34	0.059	0.412	0.529	0.265	0.735	0.022 (0.989)
	LW	24	0.916	0.042	0.042	0.938	0.063	2.023 (0.364)

**Table 2 cells-13-00720-t002:** Analysis of the effects of the different genotypes of the *PDLIM5* gene in NHP pigs.

Loci	Genotype (Sample Size)	Days to 30 kg	Days to 90 kg
−258 A > T	AA (*n* = 78)	92.15 ± 4.91 ^C^	183.53 ± 6.21 ^C^
	AT (*n* = 16)	96.36 ± 4.84 ^B^	190.99 ± 5.93 ^B^
	TT (*n* = 3)	107.07 ± 7.32 ^A^	202.93 ± 6.32 ^A^
−191 T > G	TT (*n* = 74)	91.85 ± 5.73 ^C^	183.57 ± 6.55 ^C^
	TG (*n* = 19) GG (*n* = 4)	96.26 ± 5.13 ^B^ 105.11 ± 7.45 ^A^	191.16 ± 4.62 ^B^ 201.01 ± 6.41 ^A^

Different capital letters represent significant differences among genotype groups (*p* < 0.01).

**Table 3 cells-13-00720-t003:** Joint analysis of the effects of −258 A > T and −191 T > G loci on growth traits in NHPs.

Genotype (Sample Size)	Days to 30 kg	Days to 90 kg
TG/TG (*n* = 3)	107.08 ± 7.73 ^A^	202.93 ± 6.27 ^A^
AT/AG (*n* = 9)	95.14 ± 5.49 ^B,C^	189.78 ± 6.56 ^A,B^
AT/AT (*n* = 69)	91.62 ± 5.89 ^C^	183.33 ± 5.53 ^B^
AT/TT (*n* = 5)	94.59 ± 4.61 ^B,C^	190.99 ± 7.13 ^A,B^
AT/TG (*n* = 10)	97.27 ± 4.25 ^A,B^	190.66 ± 4.82 ^A,B^
AG/TG (*n* = 1)	99.19	195.24

All sample data are expressed as mean ± SD. Statistical significances of AG/TG groups with only one individual were not analyzed.

## Data Availability

The data that support the findings of the study are available from the corresponding author upon reasonable request.

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
