# Peer review of "Expression of PDLIM5 Spliceosomes and Regulatory Functions on Myogenesis in Pigs"

_cells, 2024, doi:10.3390/cells13080720_

Round 1

Reviewer 1 Report

Comments and Suggestions for Authors

The article of Fu at al., explore the role of two isoforms of the PDLIM5 gene in the regulation of skeletal muscle development and growth in pigs. They study the expression pattern of PDLIM5short and PDLIM5long in different pig strains in order to correlate the relative presence of the two isoforms with muscle growth rate. The topic and rationale of the study is very interesting, pointing to the intriguing role of alternative splicing variants of the same gene having different effect on skeletal muscle biology. However, this relevant concept was not highlighted as it would deserve in the paper.

The authors focused on the analysis of PDLIM5 promoter region, finding interesting SNPs that regulate gene expression, nevertheless the most interesting feature of their study is the post-transcriptional regulation of the PDLIM5 gene, i.e. the mechanisms and factors that modulate the conversion from PDLIM5long to PDLIM5short though mRNA splicing. This topic was not explored nor discussed by the authors.

Given that, despite the article is undoubtedly interesting, it lacks a deep analysis of the major finding (i.e. the mechanism and function of alternative splicing). Moreover, the title itself and the conclusions do not reflect the content of the manuscript. The authors do not describe the function of PDLIM5 gene nor they identify new molecular markers for muscle growth. Moreover, they should discriminate the role of PDLIM5 in regards to proliferation, differentiation and growth of skeletal muscle. These three aspects of muscle biology are not equivalent and not interchangeable, despite linked one to the other. In particular, speaking about growth, one should distinguish between hyperplasia and hypertrophy. On this regard, the authors do not clarify whether the effect of PDLIM5short interfere with one -and which one- of these two processes.

Concluding, I could not recommend the article of Fu et al. for publication in the present form. A number of major and minor issues, besides a few typos, need to be addressed before considering it eligible of consideration. I will try to summarize them in the list here below.

Major points:

 1)Statements in the abstract (line 21) and introduction (line 83) about the role of PDLIM5 and the SNPs identified on its promoter in the control of muscle growth should be revised and scaled down in the light of the actual results, form the data of Fu e tal., PDLIM5 expression and the SNPs are only associated with the muscle growth rate. No functional data are provided on the causative action of those SNPs over muscle size.

2)Lines 35-40: the sentence is cumbersome and many repetitions are present. Moreover, I do not see the rationale of mentioning cyclins and Rb in this context. Better remove it or rephrase it.

3)The Methods section lacks the detailed description of pig muscle cell isolation and culture. Please, add the methodology for obtaining primary myoblasts culture.

4)Given the strong difference in total expression level of PDLIM5 gene in slow-growing compared to fast-growing pigs and the significant higher expression of PDLIM5long in the latter group, it would be of much interest to evaluate the relative level of short and long PDLIM5 normalized for the total amount of PDLIM5. This would enhance even more the difference in PDLIM5long expression between the pig strains and reinforce the relevance of the data. I suggest to add this piece of data. Regarding the comparison of real time PCR data to detect the relative expression level of the two isoforms (Fig. 1D), I do not agree on the representation of this result. Indeed, real time PCR assay accurately reports the relative expression level of a gene among samples, but it could not be used to test and compare the expression level of different genes despite you use the same sample and the same normalization (GAPDH in this case). This is because the efficiency and performance of cDNA amplification are different for each gene and each pair of primers. The authors should use a note of caution when considering the comparison between expression of PDLIM5long and PDLIM5short and clearly state that the result of Fig.1D is only an indication of the different relative level of expression but it should be confirm with alternative methods (droplet digital PCR or high through put RNA sequencing.

5)The text relative to Fig. 2B and 2C is misleading. Indeed, Fig. 2B shows a significant decrease of CCK8 signal in overexpressing cells, and Fig. 2C clearly shows a reduced EdU staining after PDLIM5 transfection. In the text the authors comment the Figures giving the opposite interpretation, as it is the case for the cell cycle genes qPCR data (Fig. 2D,E,F and G), these sentences should be emended. On the contrary, the last phrase (lines 193-194) is correct. In addition, it would be interesting to know the transfection efficiency of the PDLIM5 construct in pig cells (namely, how many cell are transfected in the culture?) in order to better appreciate the results in the fluorescence images in Fig.2C. The authors are invited to provide this piece of data.

6)From Fig. 3D one would say that the differentiating myoblasts are very limited in the primary culture from pig muscles. I am wondering how many muscle cells (myoblasts) are effectively present in the culture; it is well known that primary muscle culture are always contaminated by non-muscle cells such as fibroblasts, primarily, but also endothelial and others. How many non-muscle cells are present in the porcine culture ? Do they express PDLIM5? Are they transfected by Lipofectamine 3000 and thus being able to express PDLIM5short?

Minor points:

Line 15: missing “long” as part of the name of PDLIM5long

Line 17: I suggest to change the phrase “is likely to affect muscle growth…” with “is associated with a slow rate of muscle growth…”, which is actually the correct concept.

Line 52: a reference is needed.

Line 89: the author probably intended that LW pigs were introduced from abroad. Add “from”.

Lines 57-58: The fact that isoform X10 and X11 belonging to the PDLIM5short group, have shorter LIM is in contrast with the notion that PDLIM5short does not have a LIM domain. The authors should explain this apparent contradiction.

Line 67: the author should better explain the “splicing transition” occurring to the two PDLIM5 isoforms, what do they intend with “transition”?

Line 70: it is not clear what is inhibiting C2C12 proliferation and differentiation, is the PDLIM5long? This is counterintuitive since the PDLIM5long is positively associated with muscle development. Could the authors better clarify this issue?

Line 137: no definition of the acronym CCK8 is given. Every acronym should be fully described the first time it is mentioned in the text.

Line 197: “proliferating myoblasts” and not “proliferated myoblasts”. Replace “overexpression fragments”, which is not correct, with “overexpression constructs”  

Line 247: the number of repeated experiments is missing in the legend of Fig. 4

Lines 254-255: A typo should be present, since the authors wrote that the TT genotype has faster growth than TG and TT (???) genotypes, which cannot be.

Line 300: to correct the order of words: “the C2C12 myogenic…”

Line 301: remove the first “translocation” word

Comments on the Quality of English Language

The quality of English language is sufficient, no major comments on this regard.

Author Response

Response to Reviewer 1 Comments

Point 1: The article of Fu at al., explore the role of two isoforms of the PDLIM5 gene in the regulation of skeletal muscle development and growth in pigs. They study the expression pattern of PDLIM5short and PDLIM5long in different pig strains in order to correlate the relative presence of the two isoforms with muscle growth rate. The topic and rationale of the study is very interesting, pointing to the intriguing role of alternative splicing variants of the same gene having different effect on skeletal muscle biology. However, this relevant concept was not highlighted as it would deserve in the paper.

Response: Thank you for your recognition and comments on our work. We have added the descriptions in the discussion section (Lines 334-346).

Point 2: The authors focused on the analysis of PDLIM5 promoter region, finding interesting SNPs that regulate gene expression, nevertheless the most interesting feature of their study is the post-transcriptional regulation of the PDLIM5 gene, i.e. the mechanisms and factors that modulate the conversion from PDLIM5long to PDLIM5short though mRNA splicing. This topic was not explored nor discussed by the authors. Given that, despite the article is undoubtedly interesting, it lacks a deep analysis of the major finding (i.e. the mechanism and function of alternative splicing).

Response: Thank you for your suggestions. Undoubtedly, it is both interesting and worthwhile to delve into the mechanism and factors governing the mRNA splicing process that modulates the conversion from PDLIM5-long to PDLIM5-short. In this study, we mainly identified the regulatory role of PDLIM5 gene on muscle development and found that PDLIM5-short was differently expressed in the muscle of pigs with different growth rate. The expression levels of PDLIM5-long was extremely low in porcine muscle tissues. These results indicate that PDLIM5-short is the primary isoform expressed in porcine muscle tissues, and that the muscle growth rate is primarily regulated by the expression of PDLIM5-short, rather than by the conversion from PDLIM5-long to PDLIM5-short. Thus, this study focused on the role of PDLIM5-short in myogenesis and explored the regulatory expression of the PDLIM5-short, but not analyzed the mechanism of alternative splicing conversion. In the future, we plan to further investigate the mechanism and factors that modulate the isoform conversion of this gene.

Point 3: Moreover, the title itself and the conclusions do not reflect the content of the manuscript. The authors do not describe the function of PDLIM5 gene nor they identify new molecular markers for muscle growth. Moreover, they should discriminate the role of PDLIM5 in regards to proliferation, differentiation and growth of skeletal muscle. These three aspects of muscle biology are not equivalent and not interchangeable, despite linked one to the other. In particular, speaking about growth, one should distinguish between hyperplasia and hypertrophy. On this regard, the authors do not clarify whether the effect of PDLIM5short interfere with one -and which one- of these two processes.

Response: Thank you for your suggestions. We have revised the title to “Expression of PDLIM5 spliceosomes and regulatory functions on myogenesis in pigs” (Lines 1-4), and have also made corresponding modifications to the conclusions (Lines 421-423).

Point 4: Statements in the abstract (line 21) and introduction (line 83) about the role of PDLIM5 and the SNPs identified on its promoter in the control of muscle growth should be revised and scaled down in the light of the actual results, form the data of Fu e tal., PDLIM5 expression and the SNPs are only associated with the muscle growth rate. No functional data are provided on the causative action of those SNPs over muscle size.

Response: Thank you for your suggestions. We have revised the statements in the abstract (Line 25) and introduction (Lines 98-99).

Point 5: Lines 35-40: the sentence is cumbersome and many repetitions are present. Moreover, I do not see the rationale of mentioning cyclins and Rb in this context. Better remove it or rephrase it.

Response: Thank you for your suggestions. We have concisely rephrased the relevant sentences (Lines 44-46).

Point 6: The Methods section lacks the detailed description of pig muscle cell isolation and culture. Please, add the methodology for obtaining primary myoblasts culture.

Response: Thank you for your suggestions. We have added the methodology for obtaining primary myoblasts culture (Lines 158-168).

Point 7: Given the strong difference in total expression level of PDLIM5 gene in slow-growing compared to fast-growing pigs and the significant higher expression of PDLIM5long in the latter group, it would be of much interest to evaluate the relative level of short and long PDLIM5 normalized for the total amount of PDLIM5. This would enhance even more the difference in PDLIM5long expression between the pig strains and reinforce the relevance of the data. I suggest to add this piece of data. Regarding the comparison of real time PCR data to detect the relative expression level of the two isoforms (Fig. 1D), I do not agree on the representation of this result. Indeed, real time PCR assay accurately reports the relative expression level of a gene among samples, but it could not be used to test and compare the expression level of different genes despite you use the same sample and the same normalization (GAPDH in this case). This is because the efficiency and performance of cDNA amplification are different for each gene and each pair of primers. The authors should use a note of caution when considering the comparison between expression of PDLIM5long and PDLIM5short and clearly state that the result of Fig.1D is only an indication of the different relative level of expression but it should be confirm with alternative methods (droplet digital PCR or high through put RNA sequencing.

Response: Thank you for your suggestions. This study revealed that PDLIM5-short exhibited differential expression in the muscle of pigs with different growth rates, whereas the expression of PDLIM5-long was extremely low in porcine muscle tissues. These findings suggest that PDLIM5-short, as the primary isoform expressed in muscle tissues, regulates muscle growth rate in pigs. Based on this, we focused our attention on the regulatory role of PDLIM5-short in myogenesis, rather than PDLIM5-long. Consequently, we did not assess the relative level of PDLIM5-long normalized to the total amount of PDLIM5, due to its low expression levels in porcine muscle tissues and its limited relevance to our subsequent research objectives. We have revised the Figure 1 located on line 230 and rewritten the corresponding descriptions (Lines 17-22, 204-228, 231-245, 338-346).

Point 8: The text relative to Fig. 2B and 2C is misleading. Indeed, Fig. 2B shows a significant decrease of CCK8 signal in overexpressing cells, and Fig. 2C clearly shows a reduced EdU staining after PDLIM5 transfection. In the text the authors comment the Figures giving the opposite interpretation, as it is the case for the cell cycle genes qPCR data (Fig. 2D,E,F and G), these sentences should be emended. On the contrary, the last phrase (lines 193-194) is correct. In addition, it would be interesting to know the transfection efficiency of the PDLIM5 construct in pig cells (namely, how many cell are transfected in the culture?) in order to better appreciate the results in the fluorescence images in Fig.2C. The authors are invited to provide this piece of data.

Response: Thank you for your suggestions. We apologize for any inaccuracies and have revised the statements (Lines 250 and 252). In general, the longer the fragment attached to the vector, the lower the efficiency of its transfection into the cell. Therefore, to ensure high transfection efficiency, we avoid attaching additional EGFP tags, which may reduce the transfection efficiency. Without labels, it is not possible to directly distinguish transfected cells under the microscope. Nonetheless, our qPCR measurements revealed that the expression level of PDLIM5 gene was significantly increased in the overexpressed group, indicating that the majority of cells had been successfully transfected. Furthermore, this differential expression of PDLIM5 is sufficient to induce phenotypic changes, in which myoblasts proliferation and differentiation are inhibited. In addition to EdU staining, we performed marker genes expression analysis and CCK8 assays. Collectively, these results demonstrate the inhibitory effect of PDLIM5 gene on cell proliferation from various angles, thereby validating the reliability of our findings.

Point 9: From Fig. 3D one would say that the differentiating myoblasts are very limited in the primary culture from pig muscles. I am wondering how many muscle cells (myoblasts) are effectively present in the culture; it is well known that primary muscle culture are always contaminated by non-muscle cells such as fibroblasts, primarily, but also endothelial and others. How many non-muscle cells are present in the porcine culture ? Do they express PDLIM5? Are they transfected by Lipofectamine 3000 and thus being able to express PDLIM5short?

Response: Thank you for your comments. Although some other non-muscle cells may be present in the primary cells, the primary muscle cells with the largest proportion separated from the muscle will dominate the growth and gradually replace the other small number of other non-muscle cells. The following figure shows the pax7 immunofluorescence staining of the primary cells. It can be seen that the purity of the isolated primary myoblasts is relatively high. As for the poor differentiation effect, this may be caused by a variety of factors such as the different batch of differentiation medium and cell status. Even so, under the same differentiation conditions, we can still see the obvious differences in the differentiation phenotype and expression of myogenic differentiation genes between control and OE groups, which can indicate the inhibitory effect of PDLIM5 on myocyte differentiation.

IF for porcine primary myoblasts (Pax7 / DAPI)

Point 10: Line 15: missing “long” as part of the name of PDLIM5long

Response: Thank you for your comments. PDLIM5 in Line 15 indicates total PDLIM5 containing short and long isoforms. The detailed description of this results can be found in figure 1 A, B. In order to express it more clearly, we have modified the corresponding part (Line 19).

Point 11: Line 17: I suggest to change the phrase “is likely to affect muscle growth…” with “is associated with a slow rate of muscle growth…”, which is actually the correct concept.

Response: Thank you for your suggestions. We have changed the phrase “is likely to affect muscle growth…” with “is associated with a slow rate of muscle growth…” (Lines 21-22).

Point 12: Line 52: a reference is needed.

Response: Thank you for your suggestions. We have supplemented the references in line 62.

Point 13: Line 89: the author probably intended that LW pigs were introduced from abroad. Add “from”.

Response: Thank you for your suggestions. We have added “from” in line 105.

Point 14: Lines 57-58: The fact that isoform X10 and X11 belonging to the PDLIM5short group, have shorter LIM is in contrast with the notion that PDLIM5short does not have a LIM domain. The authors should explain this apparent contradiction.

Response: Thank you for your suggestions. Isoform X10 and X11 less LIM belongs to PDLIM5-short, and we have modified the descriptions (Lines 68-69).

Point 15: Line 67: the author should better explain the “splicing transition” occurring to the two PDLIM5 isoforms, what do they intend with “transition”?

Response: Thank you for your suggestions. Considering that the appearance of “splicing transition” here is inappropriate and has no practical meaning for the reading of the article, we have deleted it (Line 80).

Point 16: Line 70: it is not clear what is inhibiting C2C12 proliferation and differentiation, is the PDLIM5long? This is counterintuitive since the PDLIM5long is positively associated with muscle development. Could the authors better clarify this issue?

Response: Thank you for your suggestions. We have revised the descriptions as “The expression of the long isoform of PDLIM5 is gradually reduced during early myoblast differentiation and postnatal heart development, and it may promote the satellite cells activation and C2C12 cells myogenic differentiation by targeting the inhibitor of DNA binding 1 or 2 (Id1 or Id2)” (Lines 83-85).

Point 17: Line 137: no definition of the acronym CCK8 is given. Every acronym should be fully described the first time it is mentioned in the text.

Response: Thank you for your suggestions. We have fully described the CCK8 in line 177.

Point 18: Line 197: “proliferating myoblasts” and not “proliferated myoblasts”. Replace “overexpression fragments”, which is not correct, with “overexpression constructs” 

Response: Thank you for your suggestions. We have revised the legend of figure 2B (Lines 258-259).

Point 19: Line 247: the number of repeated experiments is missing in the legend of Fig. 4

Response: Thank you for your suggestions. We showed earlier the number of repeated experiments (n=6) in the end of the legend. In order to be consistent with the description of other legends, we have made corresponding modifications (Lines 310-311).

Point 20: Lines 254-255: A typo should be present, since the authors wrote that the TT genotype has faster growth than TG and TT (???) genotypes, which cannot be.

Response: Thank you for your suggestions. We apologize for our carelessness and have revised sentence as “The TT genotype at the-191T>G site displayed faster growth than that of the TG and TT GG genotypes (P < 0.01).” (Line 318).

Point 21: Line 300: to correct the order of words: “the C2C12 myogenic…”

Response: Thank you for your suggestions. We have corrected the order of words as “the C2C12 myogenic” (Line 384).

Point 22: Line 301: remove the first “translocation” word

Response: Thank you for your suggestions. We have removed the first “translocation” word (Line 385).

Point 23: Minor editing of English language required

Response: Thank you for your suggestions. Our manuscript was edited by an English language editing company.

Special thanks to you for your good comments.

Reviewer 2 Report

Comments and Suggestions for Authors

Dear Authors,

the general comments of your article based on the elucidation of PDLIM5 gene function in synergy with molecular markers detection for muscle growth regulation in pigs are positive.

Following some observations or doubts/curiosities.

Did your refer to PDLIM5 as candidate gene for the discovery of genotypes associated to growth traits beacuse the SNPs detection has never been conducted by now?

Why did you conduct the analysis with a coverage starting from 2000 bp upstream of the gene and not (also) on the PDLIM5 CDS region? What is the length of the entire promoter? Specifiy it in the text.

Please, add informaton and details about Sanger sequencing, including the adopted protocols and the used instruments.

Please use acronyms or for breeds or for tissues/organs because confusing in this way (i.e LW for large white; LD for longissimus dorsi).

Please uniform referring or to genoypes or to haplotypes when discuss the allele frequencies.

Lines 303-310: please better justify and discuss your "speculation" or soften it.

Best regards.

Comments on the Quality of English Language

Moderate editing of English language required.

Author Response

Response to Reviewer 2 Comments

Point 1:  Did your refer to PDLIM5 as candidate gene for the discovery of genotypes associated to growth traits because the SNPs detection has never been conducted by now?

Response: Thank you very much. There are not many papers that have reported SNPs detection of PDLIM5 so far. It was reported that polymorphisms in PDLIM5 are associated with an increased risk of dilated cardiomyopathy [1]. And the SNP of PDLIM5 (rs7690296 polymorphism) could play an important role in the pathophysiology of both bipolar disorder and schizophrenia, suggesting a potential role for PDLIM5 in the occurrence of both bipolar disorder and schizophrenia [2]. Ma J et al. assessed the genetic background and selection signatures of Huaxi cattle using high-density SNP array and speculated that the selected PDLIM5 gene may have a role in the production traits (muscle growth and differentiation) of Huaxi cattle [3]. In this study, we further identified two SNPs (-258A>T and -191T>G) in the 5’ flanking region of PDLIM5. These SNPs affected the activity of the promoter and regulated muscle growth rate in pigs. To explain the functions of these SNPs, we measured promoter activity, and the results indicated that the TG haplotype upregulated gene expression more than the AT haplotype. Furthermore, we detected the effects of different TRPC1 expressions on cellular process in porcine myoblasts. Taken together, PDLIM5 may be regarded as a candidate gene associated with growth traits.

References:

[1] Wang, D.; Fang, J.; Lv, J.; Pan, Z.; Yin, X.; Cheng, H.; Guo, X. Novel polymorphisms in PDLIM3 and PDLIM5 gene encoding Z-line proteins increase risk of idiopathic dilated cardiomyopathy. J. Cell Mol. Med. 2019, 23, 7054-7062, doi:10.1111/jcmm.14607.

[2] Zain M.A.; Roffeei S.N.; Zainal N.Z.; Kanagasundram S.; Mohamed Z. Nonsynonymous polymorphisms of the PDLIM5 gene association with the occurrence of both bipolar disorder and schizophrenia. Psychiatr Genet. 2013, 23, 258-61. doi: 10.1097/YPG.0000000000000015.

[3] Ma J.; Gao X.; Li J.; Gao H.; Wang Z.; Zhang L.; Xu L.; Gao H.; Li H.; Wang Y. et al. Assessing the genetic background and selection signatures of huaxi cattle using high-density SNP array. Animals (Basel). 2021, 11, 3469-3475. doi: 10.3390/ani11123469.

Point 2: Why did you conduct the analysis with a coverage starting from 2000 bp upstream of the gene and not (also) on the PDLIM5 CDS region? What is the length of the entire promoter? Specifiy it in the text.

Response: Thank you for your comments. In general, the sequence about 2000 bp upstream of the gene is commonly cloned as the promoter of the gene. We have added the corresponding description in lines 134 to 135. First, the upstream regions of genes contain key sequences that affect gene expression, and these SNPs may affect the binding of transcription factors, thereby altering the expression level of the gene. Therefore, screening analysis starting upstream from genes helps to identify those SNPs that may affect gene transcription and expression, contributing to a deeper understanding of gene function and regulatory mechanisms. Second, although the CDS region directly determines the amino acid sequence of proteins, not all SNPs located in the CDS region lead to changes in protein function. In contrast, SNPs in upstream regions of genes are more likely to indirectly influence protein production and thus their function by affecting transcription and expression. Thus, we conducted the analysis with a coverage starting from 2000 bp upstream of the PDLIM5 gene, and indeed find SNPs that regulate gene expression and are associated with growth traits.

Point 3: Please, add informaton and details about Sanger sequencing, including the adopted protocols and the used instruments.

Response: Thank you for your suggestions. SNPs among different pig populations were screened and identified by PCR and Sanger sequencing (ABI3030XL DNA, Thermo Fisher Scientific, Wilmington, DE, USA). Briefly, the PCR products amplified from ten individuals of each breed were pooled, and the purified products were sequenced to identify the SNPs by Sanger sequencing. Chromas Pro (Technelysium, South Brisbane, Australia) and DNAMAN6.0 (Lynnon, San Ramon, CA, USA) were used to analyze and interpret the sequencing results. To determine the genotypes of the identified SNPs, individual samples were individually used as templates for PCR amplification, and the purified products were sequenced by Sanger sequencing. Chromas Pro and DNAMAN6.0 software were used to analyze the peak situation to determine the genotype. We have added the corresponding description in the text (Lines 135-147).

Point 4: Please use acronyms or for breeds or for tissues/organs because confusing in this way (i.e LW for large white; LD for longissimus dorsi).

Response: Thank you for your suggestions. We have modified the legend of figure 1 in the way you suggested (Lines 235-236).

Point 5: Please uniform referring or to genotypes or to haplotypes when discuss the allele frequencies.

Response: Thank you for your suggestions. We have uniformly referred to genotypes when discuss the allele frequencies (Lines 412-414).

Point 6: Lines 303-310: please better justify and discuss your "speculation" or soften it.

Response: Thank you very much for your suggestions. Our previous statement indeed appeared to be unscientific and not rigorous enough. And we have softened it and modified it in lines 387-398.

Point 7: Moderate editing of English language required

Response: Thank you for your suggestions. Our manuscript was edited by an English language editing company.

Special thanks to you for your good comments.

Reviewer 3 Report

Comments and Suggestions for Authors

Dear Authors, 

The manuscript under the title:" Identifying the Function of PDLIM5 and Molecular Markers for Muscle Growth Regulation in Pigs"  is interesting and explores novel polymorphisms within the promoter of the PDLIM5 gene, its potential impact on gene function, transcript variants and association with muscle growth rate. The manuscript is well-written, and the methods are properly described and used. My biggest concerns are:

1.  In ln 90-92, you stated "embryonic tissues" and next LD muscle tissues were sampled. This should be clarified: numbers, fetuses, adults. Exactly: what and from and when.  And what samples you were using for qPCR .

2. If you use embryonic tissues this means that you ending up pregnancies if so -in my opinion,  the proper ethical statement should be included in this section. 

3. For association study the numbers of genotypes are quite low. 

the minor comments:

ln 90-94: Please clarify the numbers of samples and time points of sampling.

ln 101 : (sqRT-PCR) s missing

Author Response

Response to Reviewer 3 Comments

Point 1: In 90-92, you stated "embryonic tissues" and next LD muscle tissues were sampled. This should be clarified: numbers, fetuses, adults. Exactly: what and from and when.  And what samples you were using for qPCR.

Response: Thank you for your suggestions. For RNA extraction, six embryonic tissues from each group (TP, WJ and LW) were sampled from two pregnant sows after 60 days of insemination. These embryonic tissues specifically included heart, back fat (BF), lung, liver, leg and intestines, all of which were used for sqRT-PCR. Additionally, embryonic longissimus dorsi (LD) muscle tissues taken from the 12th rib were sampled for both sqRT-PCR and qRT-PCR. We have added the description to the revised manuscript (Lines 109-114).

Point 2: If you use embryonic tissues this means that you ending up pregnancies if so -in my opinion,  the proper ethical statement should be included in this section.

Response: Thank you for your suggestions. The animal feeding and tests were conducted based on the National Research Council Guide for the Care and Use of Laboratory Animals and were approved by the Institutional Animal Care and Use Committee (approval number: AW80203202-1-1). We have included the corresponding description in the revised manuscript (Lines 118-121).

Point 3: For association study the numbers of genotypes are quite low.

Response: Thank you for your comments. We have made every effort to collect as many NHPs with phenotypic data records and have analyzed the associations between them and growth trait . However, in recent years, the African swine fever has had a significant impact, limiting the number of pig populations for which phenotypic data can be obtained. Although some individuals with genotypes such as AG/TG comprised a relatively small proportion of the population, the overall sample size was still sufficient to demonstrate a correlation between SNPs and growth traits to a certain extent. Of course, if conditions allow in the future, we can further increase the population size or even expand our study to other populations to verify the correlation between these two SNPs and growth traits. This will be beneficial for the better application of molecular markers in genetic breeding.

Point 4: ln 90-94: Please clarify the numbers of samples and time points of sampling.

Response: Thank you for your suggestions. For RNA extraction, at least six embryos from each group (TP, WJ and LW) were sampled from two pregnant sows after 60 days of insemination. These embryonic tissues specifically included heart, back fat (BF), lung, liver, leg and intestines, all of which were used for sqRT-PCR. Additionally, embryonic longissimus dorsi (LD) muscle tissues taken from the 12th rib were sampled for both sqRT-PCR and qRT-PCR. We have added the description to the revised manuscript (Lines 109-114).

Point 5: ln 101: (sqRT-PCR) s missing

Response: Thank you for your careful reminder. We have modified it in line 127.

Special thanks to you for your good comments.

Reviewer 4 Report

Comments and Suggestions for Authors

This manuscript entitled "Identifying the Function of PDLIM5 and Molecular Markers for Muscle Growth Regulation in Pigs" by Yu Fu et al. identified the expression of PDLIM5-short across various swine species with differing growth characteristics, identifying two single nucleotide polymorphisms as potential biomarkers for pig breeding. For mechanistic studies, the authors reported that PDLIM5-short inhibits myoblast proliferation through the regulation of CCND1, Ki67, PCNA, and CDK4. Additionally, they presented results indicating that PDLIM5 plays a negative role in myogenic differentiation by inhibiting MyoG, MyoD, and MyHC.

This is an interesting study. The manuscript is well-written. The reviewer has only some minor concerns as follows:

1. A weak point of this study is the sample size of some swine species for SNP assay. However, it can be accepted.

2. In the Materials and Methods section, the authors could include information about the gender of the animals used.

3. While PDLIM5-short inhibits myoblast proliferation through CCND1 regulation, interestingly, there was no significant decrease observed in CCND1 levels after overexpressing PDLIM5-short in porcine primary myoblasts. It may be beneficial to provide further description to elucidate this observation.

Comments on the Quality of English Language

Minor editing of English language required.

Author Response

Response to Reviewer 4 Comments

Point 1: A weak point of this study is the sample size of some swine species for SNP assay. However, it can be accepted.

Response: Thank you for your comments and understanding. We have tried our best to collect as many samples size of swine species as possible. However, in recent years, due to the impact of African swine fever, the number of pig populations can be obtained is limited. Although the sample size of some swine species is not very sufficient, the overall sample size can still explain the correlation between SNPs and growth traits to a certain extent. Of course, if conditions permit in the future, we can further expand the population number or even include other more populations to verify the SNPs, which will be conducive to the better application of molecular markers in genetic breeding.

Point 2: In the Materials and Methods section, the authors could include information about the gender of the animals used.

Response: Thank you for your suggestions. There is a certain relationship between pig muscle development and sex. In our study, the muscle tissues were obtained from embryos at approximately 60 days of embryonic age. Since the influence of gender on embryonic muscle development is minimal and the number of embryonic samples was small, we did not distinguish between genders when sampling for gene expression tests. As for SNPs, our focus was primarily on single nucleotide polymorphisms between different populations, not genders, so we did not consider the genders of pigs. For correlation analysis between genotypes and phenotypes, 97 boars from the NHP group were used. We have added this information in the revised manuscript (Line 148).

Point 3: While PDLIM5-short inhibits myoblast proliferation through CCND1 regulation, interestingly, there was no significant decrease observed in CCND1 levels after overexpressing PDLIM5-short in porcine primary myoblasts. It may be beneficial to provide further description to elucidate this observation.

Response: Thank you for your comments. Overexpression of PDLIM5-short inhibits myoblast proliferation but does not significantly regulate the expression of CCND1, possibly due to multiple complex factors. Firstly, the expression of CCND1 is not solely controlled by a single gene but is part of a complex network involving multiple upstream genes and transcription factors. Even if a gene promoting cell proliferation is overexpressed, its effects may be counteracted or modified by other regulatory factors, resulting in no significant change in CCND1 expression. Secondly, gene expression is regulated at multiple levels, including transcriptional regulation, post-transcriptional modification, protein translation and modification. Therefore, even if a gene is overexpressed at the transcriptional level, its ultimate impact on CCND1 expression may be limited by other regulatory mechanisms. Furthermore, intracellular gene expression is a dynamic equilibrium process with intricate interactions and feedback mechanisms among various genes and proteins. Overexpression of a gene may trigger a cascade of reactions that may not directly correlate with CCND1 expression. We have provided further discussion to elaborate on this observation (Lines 356-369).

Point 4: Minor editing of English language required

Response: Thank you for your suggestions. Our manuscript was edited by an English language editing company.

Special thanks to you for your good comments.

Round 2

Reviewer 1 Report

Comments and Suggestions for Authors

The authors carefully considered all my comments and followed the proposed suggestions, answering to all the questions arisen. I am quite satisfied of the work they have done in order to reply to my points. The revised version of Fu at al. encloses indeed many modifications and improvements. However, I still need to highlight a couple of additional issues that I recommend the authors to take into consideration.

-          The Figure of Pax7 staining provided in the rebuttal letter to testify the homogenous presence of myogenic cells in the porcine primary culture is not convincing. Indeed, Pax7 antibodies should react with the transcription factor in the nucleus resulting in a specific nuclear staining. In the image obtained by Fu et al, the green signal (Pax7) is mostly aspecifically marking the cell cytosol. Only few cells show a faint nuclear staining, which is in line with my comment that a limited myogenic cells are present in the primary culture obtained by developing pig muscles. Given the fact that Pax7 antibodies are not normally super-efficient and that Pax7 expression rapidly turn off upon culturing of muscle cells, MyoD staining would have been a better choice.

-          Line 25: “promotor” is not correct, substitute with “promoter”

Considering that a major revision of the manuscript has been accomplished and the changes I recommended were introduced, I am pleased to evaluate the work of Fu et al. acceptable for publication in Cells.

Author Response

Response to Reviewer 1 Comments

The authors carefully considered all my comments and followed the proposed suggestions, answering to all the questions arisen. I am quite satisfied of the work they have done in order to reply to my points. The revised version of Fu at al. encloses indeed many modifications and improvements. However, I still need to highlight a couple of additional issues that I recommend the authors to take into consideration.

Point 1: The Figure of Pax7 staining provided in the rebuttal letter to testify the homogenous presence of myogenic cells in the porcine primary culture is not convincing. Indeed, Pax7 antibodies should react with the transcription factor in the nucleus resulting in a specific nuclear staining. In the image obtained by Fu et al, the green signal (Pax7) is mostly aspecifically marking the cell cytosol. Only few cells show a faint nuclear staining, which is in line with my comment that a limited myogenic cells are present in the primary culture obtained by developing pig muscles. Given the fact that Pax7 antibodies are not normally super-efficient and that Pax7 expression rapidly turn off upon culturing of muscle cells, MyoD staining would have been a better choice.

Response: Thank you for your suggestions. We are sorry that we do not have MyoD antibodies that can be used for immunofluorescence staining at present. Considering that it takes a long time to order new antibodies, we used another skeletal muscle-specific expression antibody (α-actin) to identify the isolated porcine skeletal muscle primary cells ( shown below).

Point 2: Line 25: “promotor” is not correct, substitute with “promoter”

Response: Thank you for your suggestions. We have revised the spelling mistake in Line 25.

Point 3: Considering that a major revision of the manuscript has been accomplished and the changes I recommended were introduced, I am pleased to evaluate the work of Fu et al. acceptable for publication in Cells.

Response: Special thanks to you for your good comments.

Reviewer 2 Report

Comments and Suggestions for Authors

The authors amended and modified the manuscript according to suggestions. 

Comments on the Quality of English Language

Minor editing of English is required.

Author Response

Response to Reviewer 2 Comments

Point 1:  The authors amended and modified the manuscript according to suggestions. Minor editing of English is required.

Response: Thank you for your suggestions. Our manuscript was edited by an English language editing company.

Special thanks to you for your good comments.

Reviewer 3 Report

Comments and Suggestions for Authors

Thank you for applying my suggestions and clarifying my concerns. 

From my point of view, the manuscript is suitable for publishing.

Author Response

Response to Reviewer 3 Comments

Thank you for applying my suggestions and clarifying my concerns.

From my point of view, the manuscript is suitable for publishing.

Response: Special thanks to you for your good comments.

Reviewer 4 Report

Comments and Suggestions for Authors

I do not have further questions.

Author Response

Response to Reviewer 4 Comments

I do not have further questions.

Response: Special thanks to you for your good comments.